# A Simple and Provable Approach for Learning on Noisy Labeled Medical Images

### Nan Wang
East China University of Science and
Technology
Shanghai, China
WangN@ecust.edu.cn

### Zonglin Di
University of California, Santa Cruz
CA, USA
zdi@ucsc.edu

### Houlin He
University of British Columbia
Vancouver, Canada
hehoulin123@gmail.com

### Qingchao Jiang*
Key Laboratory of Smart
Manufacturing in Energy Chemical
Process, Ministry of Education
East China University of Science and
Technology
Shanghai, China
qchjiang@ecust.edu.cn

### Xiaoxiao Li*
University of British Columbia
Vancouver, Canada
xiaoxiao.li@ece.ubc.ca

## Abstract

Deep learning for medical image classification needs large amounts of carefully labeled data with the aid of domain experts. However, data labeling is vulnerable to noises, which may degrade the accuracy of classifiers. Given the cost of medical data collection and annotation, it is highly desirable for methods that can effectively utilize noisy labeled data. In addition, efficiency and universality are essential for noisy label training, which requires further research. To address the lack of high-quality labeled medical data and meet algorithm efficiency requirements for clinical application, we propose a simple yet effective approach for multi-field medical images to utilize noisy data, named `Pseudo-T correction`. Specifically, we design a noisy label filter to divide the training data into clean and noisy samples. Then, we estimate a transition matrix that corrects model predictions based on the partitions of clean and noisy data samples. However, if the model overfits noisy data, noisy samples become more difficult to detect in the filtering step, resulting in inaccurate transition matrix estimation. Therefore, we employ gradient disparity as an effective criterion to decide whether or not to refine the transition matrix in the model's further training steps. The novel design enables us to build more accurate machine-learning models by leveraging noisy labels. We demonstrate that our method outperforms the state-of-the-art methods on three public medical datasets and achieves superior computational efficiency over the alternatives.

---

*Corresponding authors

## CCS Concepts

• **Computing methodologies → Machine learning**.

## Keywords

Noisy Label Learning, Multi-field Data, Medical Image Analysis, Pseudo-label

**ACM Reference Format:**
Nan Wang, Zonglin Di, Houlin He, Qingchao Jiang, and Xiaoxiao Li. 2024. A Simple and Provable Approach for Learning on Noisy Labeled Medical Images. In *Proceedings of the 32nd ACM International Conference on Multimedia (MM '24), October 28-November 1, 2024, Melbourne, VIC, Australia.* , 9 pages. https://doi.org/10.1145/3664647.3681274

## 1 Introduction

Medical image classification plays a vital role in clinical treatment tasks and diagnosis applications [29, 38, 39]. The success of these classification tasks is contingent upon learning highly discriminative representations using deep neural networks from a huge amount of carefully labeled data with the assistance of domain experts. However, labeling samples by medical experts for medical image classification is expensive, and collecting labels may suffer from the noises [17]. For instance, as demonstrated in Fig. 1, the malignant histology lymph node images exhibit colors and structures that are strikingly similar to benign ones. This indicates that noisy annotations are an unavoidable part of medical image processing in real-world settings.

With the development of deep learning, various methods have been widely used in medical image classification [3, 6, 28] to solve the issue of noisy labeling. There exist two general resolving directions, one is to collect more data [7, 34], although data collection methods are typically affected by noisy labels. Another way is to fully utilize clean labels while ignoring the noisy labels [11, 12, 18, 36]. Despite making better annotations, these methods are over-complex in the architecture design compared with vanilla classification training, thus becoming sensitive to outliers. Equally, simply and provably processing noisy labeled medical data

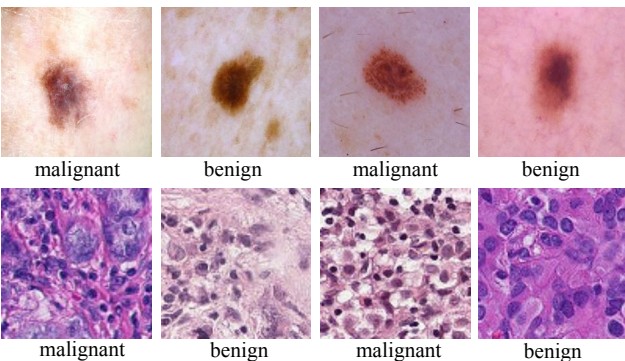

| malignant | benign | malignant | benign |

| malignant | benign | malignant | benign |

**Figure 1: Examples of typical confusing malignant and benign images from skin lesions and histology lymph nodes, which are prone to inaccurate labeling during the practical annotation process.**

is significant to assist doctors in diagnosis, few researches have been conducted in medical image classification. Specifically, the existing state-of-the-art method RMIC [38] introduces a complicated teacher-student model that has two carefully designed distillation modules. This is problematic as it costs more resources and time to yield a model. Though the issue can be overcome by optimizing the speed and enhancing the hardware configuration, it is still better to have a lightweight method without sacrificing prediction accuracy. Improving model efficiency is equally important as maintaining model accuracy. Apparently, many clinical applications require real-time decision [23] to allow human-in-the-loop, while a complex model cannot meet such a requirement. Besides, sophisticated deep learning models for healthcare are challenging to install and update on power-constrained devices [21]. Additionally, it is critical to employ simple noisy label training, while the correctness is provable. Such methods have merely been studied so far in medical image classification.

Despite their extraordinary effectiveness, it has been repeatedly observed that the performance of these methods is easily influenced by the bias of the training set in complex real-world scenarios [38]. For instance, such bias could come from the variations of disease patterns. A typical issue is that these data biases can cause noise labeling problems. Therefore, an important topic in the realm of medical image analysis is handling multimodal noise. As far as we know, recent research has used cross-modal weak supervision by utilizing convolutional neural networks (CNN). While various previous cross-modal weak supervision methods have shown promise in medical applications, the research conducted to date has been on generating weak labels in an application-specific manner [5]. There is no theoretical analysis that describes the performance expectations of the resulting models, which would support any of these application-specific cross-modal weak supervision approaches. Additionally, these methods can only handle MRI or CT, and the generalizability needs to be improved.

By analyzing the existing noisy label learning work for medical image classification, we find that most of the existing methods focus on detecting and utilizing single-mode clean labels only. Nevertheless, these methods are heuristic without provable properties and

require much more training parameters than vanilla classification model [18, 21, 23]. Noticing the gaps, we contribute as follows:

- We propose a simple and provable noisy label learning strategy that can leverage the labels of clean and noisy data. Specifically, we estimate a transition matrix to correct model outputs to fit the observed labels.
- We theoretically prove that our method achieves the same optimal as learning with all clean labels on expectation. To overcome the model overfitting noisy data, we employ gradient disparity as a criterion so that the fixed transition matrix can be accurately modeled by exploiting the noise.
- Extensive experiments demonstrate the superior performance of the proposed framework. Our method also has been extensively validated on three public medical datasets with noisy labels, notably outperforming state-of-the-art methods.

## 2 Related Work

### 2.1 Deep Learning With Noisy Labels

To mitigate the impacts of noisy labels on deep learning, researchers proposed approaches to increase the deep neural networks' robustness in recent years.

Attempts have been made to evaluate the training labels' correctness. The concept of a supervisor model, MentorNet, is introduced by Jiang et al. [11]. MentorNet is initialized to approximate a predefined sample weighting scheme. During training, it learns with the base deep learning network and re-weights the sample labels based on their reliability. However, the ubiquity of ambiguous hard samples makes the method difficult for medical images, since the images of two classes can be pretty similar. Other than that, other approaches tend to gradually correct the target labels according to the model's prediction. Reed et al. [27] proposed the bootstrapping method, considering the training label confidence by evaluating the model prediction consistency on similar input data. Instead of trusting all training labels to be ground truth, the model's current prediction also plays an important role in deciding the model's label targets. Based on the bootstrapping approach, Iscen et al. [10] penalizes prediction divergence between samples with similar features via an additional consistency loss layer. However, this method introduces additional computational costs to determine the similarity and weighted predictions. It is usually costly for medical image datasets, which are usually large in image feature dimensions. Other than that, the samples with noisy labels are not utilized.

Meanwhile, some methods increase the training loss robustness to noisy labels. Sukhbaatar et al. [30] allows the model to capture the label transition matrix by introducing a constrained linear noise layer, handling both label flips and outliers. Patrini et al. [25] proposed two loss correction procedures, which are independent of the neural network's application and architecture. With an assumption of possessing a known label transition matrix $T$, the loss can be corrected by multiplying with the inverse of $T$. However, these methods require a certain amount of clean data ahead of time, and the transition matrix is hard to capture due to the complexity of medical image data. Being independent of any prior information, [13] filters noisy labels and trains on the labels with high confidence by selectively applying negative and positive learning.

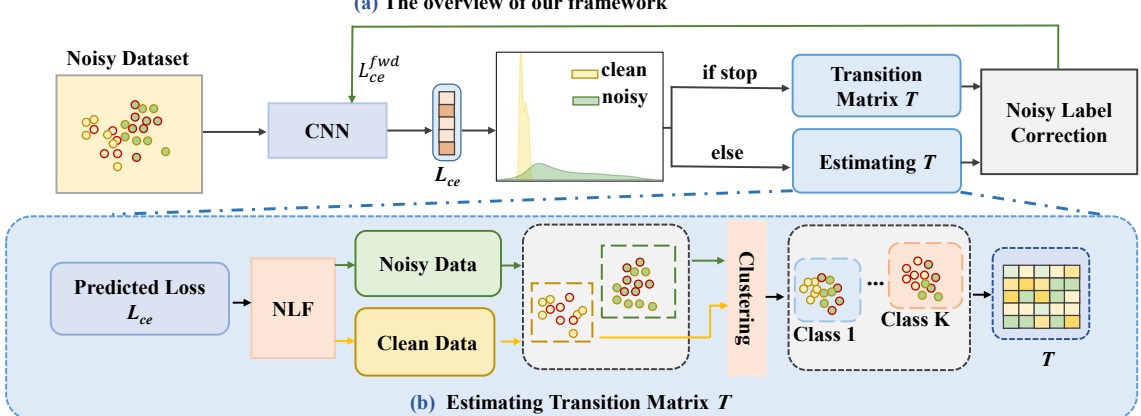

**Figure 2: The proposed `Pseudo-T correction` method, including (a) the overview of our framework, (b) indicates the specific procedure of estimating the transition matrix. We first input the noisy dataset into our network and divide all samples into noisy and clean groups. To tackle the challenge of noisy label detection, a noisy label filter (NLF) divides all the training data by leveraging the predicted cross-entropy loss of our network. Noisy data are framed by red lines, and their labels need to be corrected. Then, we design a transition matrix that corrects model predictions based on the partitions of clean and noisy data samples. To prevent overfitting of the model to noisy data, we utilize gradient disparity (GD) as a stopping criterion to determine whether to fix the transition matrix $T$ in further model training steps.**

## 2.2 Noisy Labels in Medical Imaging

Labeling medical images is expensive and time-consuming as it requires human experts to manually classify them. The data source can also inevitably suffer from noises [17]. Some ambiguous images may confuse exports and some large-scale datasets require automation tools to extract labels, resulting in a certain level of wrong annotation [38]. To address such an issue, many researches were conducted.

Some researchers focus on identifying noisy data from clean datasets. Dgani et al. [4] trains a stochastic matrix to determine the likelihood that a label is noisy. Zhu et al. [41] proposes a label quality evaluation criteria to only feed preferable data to optimize the network's parameters. Kim et al. [12] developed the FINE framework to filter out noisy data based on their eigenvectors.

Rather than implementing a noisy data filter, some approaches also consider utilizing them to enhance feature detection, and meanwhile, suppressing their influence by re-weighting. Xue et al. [37] designs an iterative learning framework to identify wrongly annotated data and hard samples. A module is also developed to consider the usefulness of those data by assigning appropriate weights. Xue et al. [38] introduces the RMIC method, which involves a co-training framework. The framework includes two student-teacher models, which classify clean and noisy labels independently. Their classifications are fed to one another as new inputs in the next training cycle. Within a student-teacher network, the clean and noisy data are respectively considered by four losses. To increase the teacher network's robustness to noisy labels, its weight is updated by the student network's exponential moving average. The noisy labels are also utilized by a self-supervising learning strategy to improve the network's feature learning. Liu et al. [18] addresses the "noisy supervised domain adaptation" problem for medical imaging segmentation with S-CUDA, which also contains two peer networks.

The high-confident noisy labels are identified and corrected by the network's predictions for future reuse.

## 2.3 K-Mean Clustering

K-mean clustering is one of the most widely used data mining algorithms, grouping up samples with the closest mean. In recent years, many efforts have been made to improve the K-mean algorithm efficiency. Leng et al. [15] chooses cluster initial points by comparing the data point distances with a small threshold. It refines the clusters and decides whether to merge them with an influence factor to measure the similarities between the two clusters. Napoleon and Lakshmi [22] proposes a more efficient K-mean clustering approach of two phases. The clustering centroids are initialized in the first phase, and data points are assigned and clusters are finalized in the second phase. Rauf et al. [26] further improves the clustering efficiency by first classifying data into K arrays of the same size. The clusters are then finalized in the second phase by computing and comparing the distance between data points to the cluster means.

## 3 Preliminary

### 3.1 Problem Setting

We denote $x \in X$ as an image sample, $\tilde{y} \in \{e^i | i \in [K]\}$ is its observed label but possibly noisy (*i.e.*, incorrect), a.k.a corrupted label. Here, $K$ is the number of classes, $e^i$ is a $K$-dimensional one-hot vector, whose $i$-th entry is 1 and else are 0s. The ground truth label $y \in \mathcal{Y}$ of the sample is unknown. Under these setting, we train a classifier $f_\theta$ to fit the input images $X$ to their noisy labels $\tilde{\mathcal{Y}}$, we aim to optimize the our classifier $f_\theta$ with $\{X, \tilde{\mathcal{Y}}\}$. The expected risk is defined as

$$\mathcal{R}(f_\theta) = \mathbb{E}_{(x,y) \sim P_{X,\tilde{y}}}[\mathcal{L}(\hat{f}_\theta(x), \tilde{y})], \quad (1)$$

where $\mathcal{L}$ is cross-entropy loss, and we aim to find $\theta^* \coloneqq \arg\min \mathcal{R}(f_\theta)$.

***Label Noise.*** The process of label corruption can be represented by randomly flipping a true label $y$ to $\tilde{y}$ with probability $p(\tilde{y}|y)$. Thus, we observe samples from distribution $p(x, y)$, where $p(x, y) = \sum_y p(\tilde{y}|y)p(y|x)p(x)$. Such a transition can be expressed as a Markov transition matrix $T \in [0, 1]^{K \times K}$, with $T_{ij} \coloneqq p(\tilde{y} = e^j|y = e^i)$, such that $p_{\tilde{y}|x} = T^\top p_{y|x}$.

## 3.2 Forward Correction in Learning on Label Noise

Risk correction is a simple yet effective approach in learning on label noise [25, 30, 31], among which forward correction dominates the performance in practice [25]. Without risk correction, training a neural network results in a prediction for the corrupted labels $\hat{p}(\tilde{y}|x)$. Forward correction corrects the model prediction using transition matrix $T$. For classification based on minimizing cross-entropy, we have the forward corrected loss:

$$\mathcal{L}_{ce}^{fwd}(e^i, \hat{p}(y|x)) = -\log \hat{p}(\tilde{y} = e^i|x) \tag{2}$$

$$= -\log \sum_{j=1}^K p(\tilde{y} = e^i|y = e^j)\hat{p}(y = e^j|x) \tag{3}$$

$$= -log \sum_{j=1}^K T_{ij}\hat{p}(y = e^j|x). \tag{4}$$

Obviously, forward correction simply replaces the prediction trained on the noisy label $\hat{p}(y = e^j|x)$ with $T_{ij}\hat{p}(y = e^j|x)$.

***Optimal in Forward Correction.*** In our context, the model is optimal in the sense that the minimizer of the corrected loss task coincides with the minimizer of the supervised task on true labels as if the perfect true labels $y$ can be observed.

THEOREM 3.1. *(Equal optimal [25])*
*Given the forward corrected loss defined in Eq. 4 and $T$ is non-singular, the minimizer of the corrected loss under the noisy label distribution is the same as the minimizer of the original loss under the true label distribution:*

$$\arg\min_f \mathbb{E}_{p(x,\tilde{y})}\left[\mathcal{L}_{ce}^{fwd}(Tf(x), \tilde{y})\right]$$
$$= \arg\min_f \mathbb{E}_{p(x,y)}\left[\mathcal{L}_{ce}(f(x), y)\right] \tag{5}$$

PROOF. Consider a link function $\psi : \Delta^{K-1} \mapsto \mathbb{R}^c$, which is invertible, the softmax function is the inverse link function in cross-entropy loss $\mathcal{L}_{ce} = -\log p(y = e^i)_i = -\log \frac{\exp(W_{i\cdot}^{(n)} x^{(n-1)+b_i^{(n)}})}{\sum_{k=1}^c \exp(W_{k\cdot}^{(n)} x^{(n-1)+b_k^{(n)}})}$, where $n$ is the index of layers. Let $h(\cdot)$ represent the features that pass through the softmax function. With proper composite loss properties of cross-entropy, the minimizer satisfies

$$h^* = \arg\min_h \mathbb{E}_{x,y}\mathcal{L}_\psi(y, h(x)) = \psi(p(y|x)), \tag{6}$$

---

**Algorithm 1** The training process of `Pseudo-T correction`.

**Input:** Training data $(x, y) \in (\mathcal{X}, \mathcal{Y})$, network $g(x; \theta)$, transition Matrix $T$, Epoch $\mathcal{T}$.

1: **repeat**
2:     **for** $t \leftarrow 1$ to $\mathcal{T}$ **do**
3: Batch $\leftarrow$ Sample $x$
4: Update $g$ by minimizing Eq. 4
5: Compute the cross-entropy loss $\mathcal{L}$
6:     **if** GD =! best GD
7: Clean & Noisy Sample $x \leftarrow$ NLF (Sec. 4.2.1)
8: update transition Matrix $T$
9:     **do**
10: $T_{ij} = p(\tilde{y} = e^j|y = e^i)$
11:     **else**
12: fix transition Matrix $T$
13:     Update label probabilistic modeling (Sec. 4.3)
14: **until** Convergence or $\mathcal{T}$ reach maximum iterations.

---

The proof follows [25]. Notice that:

$$\mathcal{L}_{ce}^{fwd}(y = e^i|x) \tag{7}$$

$$= \mathcal{L}_{ce}(y, T\psi^{-1}(h(x))) \tag{8}$$

$$= \mathcal{L}_{ce}^\phi(y, h(x)), \tag{9}$$

where $\phi^{-1} = \psi^{-1} \circ T$ and $\phi = (T^{-1}) \circ \psi$. Eq. 9 is the proper composite with link $\phi$. Finally, from Eq. 9, the loss minimizer is

$$h^* = \arg\min_h \mathbb{E}_{x,\tilde{y}}\mathcal{L}_\phi(y, h(x))$$
$$= \phi(p(\tilde{y}|x))$$
$$= \psi(T^{-1}p(\tilde{y}|x))$$
$$= \psi(p(y|x))$$

$\square$

## 4 Equations
## 4.1 Overview of the Framework

The goal of our work is to design a simple method for medical image analysis (see Fig. 2), while its correctness is provable. The overall training pipeline of our method is still described in Algorithm 1, Motivated by the essential theoretical guarantee in Theorem 3.1, we focus on applying the simple loss forward correction to medical tasks. However, the existing related works [25] typically assume the transition matrix $T$ is known, which is impractical in medical applications. Medical image annotation contains complex human decisions, thus the real $T$ is difficult to model. To estimate transition matrix $T$ for forwarding correction, we first use label prediction by noisy label filter (NLF) (see Section 4.2.1) to distinguish clean samples from noisy samples that serve as the input for training the network. Then, the gradient disparity is applied as a stopping criterion to determine if the matrix is fixed. Finally, the transition matrix is applied to give different confidences to the clean and noisy labels, adapted to the noisy (see Section 4.2).

## 4.2 Estimating Transition Matrix $T$

The main purpose of this work is to estimate the transition matrix $T$ for forward correction described in Section 3.2. To this end, the key step is to estimate the probability of a sample in class $i$ is labeled as $j$, $T_{ij} = p(\tilde{\boldsymbol{y}} = \boldsymbol{e}^j | \boldsymbol{y} = \boldsymbol{e}^i)$. We propose to estimate the $T_{ij}$ from the sample distribution of the training dataset. Given that corrupted labels $\tilde{\boldsymbol{y}} \in \tilde{\mathcal{Y}}$ are observed for $N$ training samples if their corresponding true labels were known, we can count the number of samples $N_i$ in class $i$. Hereby, the estimated transition matrix is expressed as $T_{ij} = p(\tilde{\boldsymbol{y}} = \boldsymbol{e}^j | \boldsymbol{y} = \boldsymbol{e}^i) \simeq \frac{|\{\tilde{\boldsymbol{y}}_n = j | n \in [N_i]\}|}{N_i}$ for $i, j \in K$, where $K$ is the number of classes.

*4.2.1 Label Prediction by Noisy Label Filter (NLF).* Patrini [25] assumes that the transition matrix $T$ is known ahead of time. However, it is very hard to estimate such a matrix due to the complexity of medical images. To estimate transition matrix $T$, we first filter out clean data, which are believed to be correctly labeled in the dataset. Inspired by Xue et al. [38], we use an NLF to separate clean and noisy labels (a.k.a. wrong labels) based on data point-wise classification loss. More specifically, we compute the cross-entropy loss $\mathcal{L}(\hat{\mathcal{Y}}, \tilde{\mathcal{Y}})$ at the end of every epoch, where $\hat{\mathcal{Y}}$ is the prediction result/are the prediction results from $f_\theta(\mathcal{X})$. In our experiments, we use the cross-entropy loss as $\mathcal{L}$. Next, we input the loss into a noisy label filter (NLF) to predict $\{\mathcal{X}, \tilde{\mathcal{Y}}\}$ into clear $\{\mathcal{X}_c, \tilde{\mathcal{Y}}_c\}$ and noisy $\{\mathcal{X}_n, \tilde{\mathcal{Y}}_n\}$ samples. In detail, the NLF employs a two-component Gaussian Mixture Model (GMM) to cluster the samples into two groups based on their point-wise cross-entropy losses (*i.e.,* , $\ell_i = \hat{p}(\tilde{\boldsymbol{y}}_n | \boldsymbol{x}_n)$) for data point $n$.

The NLF method only selects the most confident samples as clean samples at the initializing stage, and the confidence threshold gradually decreases as the model becomes more robust. With a high confidence threshold at the beginning, the consistency of the transition matrix $T$ is maintained to the greatest extent. Meanwhile, we added gradient disparity (GD) as a stopping criterion to determine whether or not we should stop updating the transition matrix T.

*4.2.2 Estimate $N_i$ Via Clustering.* The key challenge exposed is the true label being unknown, thus the number of samples $N_i$ within a given true label $i$ are unavailable. To tackle this issue, we propose an unsupervised pseudo-labeling strategy to estimate $N_i$. Specifically, we warm up the model without correction for $\tau$ steps and then perform clustering on the data feature representations to form $K$ clusters, then assign each cluster a label based on the majority voting on a subset of clean samples within each cluster.

We first extract informative low-dimensional representations from the input medical images. Convolution neural networks (CNNs) show their efficiency in extracting the feature. Further, a recent work [16] shows that extracted features are representative and robust even when being trained on label noise. In our method, we use a ResNet-18 [9] as the feature extractor and one linear layer as the classifier training on the observed noisy labels. Then, we perform K-means clustering [26] on the extracted features to split the training data into $K$ clusters.

*4.2.3 Assign Pseudo Labels to Clusters.* Towards estimating $N_i$, one step further is to assign a pseudo label to each cluster. For each cluster $C_i, 0 \le i < K$, we split the data samples $(\mathcal{X}^{C_i}, \mathcal{Y}^{C_i})$

in cluster $C_i$ into clean $(\mathcal{X}_c^{C_i}, \tilde{\mathcal{Y}}_c^{C_i})$ (correctly labeled) and noisy $(\mathcal{X}_c^{C_i}, \tilde{\mathcal{Y}}_n^{C_i})$ (wrongly labeled) samples using the NLF method given in Section 4.2.1.

The label of each cluster depends on the majority voting on the prediction of clean data. To achieve more reliable pseudo labeling, we only choose the top 60% of the clean data given the noise level in medical data annotation is typically small high (*i.e.,* $\le 40\%$) for mapping each $C_i$ to a non-overlapped pseudo-class.

*4.2.4 Estimate $T_{ij}$ via Counting.* After labeling each cluster, we denote the number of each class $j \in [K]$ in cluster $C_i$ with pseudo true label $i$ as $c_i^j = |\{\tilde{\boldsymbol{y}}_n = j | n \in |C_i|\}|$ and $\sum_{j=1}^K c_i^j = |\tilde{\mathcal{Y}}_c^{C_i}|$. Thus, $c_i^j$ is achieved by counting the number of samples in each class based on the corrupted labels $\tilde{\boldsymbol{y}} \in \tilde{\mathcal{Y}}$. For $i, j \in K$, we have the estimation of transition matrix $T$ written as

$$T_{ij} = p(\tilde{\boldsymbol{y}} = \boldsymbol{e}^j | \boldsymbol{y} = \boldsymbol{e}^i) \simeq \frac{|\{\tilde{\boldsymbol{y}}_n = j | n \in [N_i]\}|}{N_i} \simeq \frac{c_i^j}{|C_i|}. \quad (10)$$

For Estimate $T_{ij}$, the model will learn the clean data in the early epochs based on the memorization effect. Thus, the model tends to find the noisy label and predict the correct transition matrix $T$. Next, the loss from the clean and noisy labels can differ greatly. The NLF can filter the noisy ones. Furthermore, the clustering can distinguish between clean and noisy pairs. These three techniques can help the estimated transition matrix converge to the real one.

## 4.3 Label Probabilistic Modeling and Updating

After we have estimated the transition matrix $T$, we apply the forward correction to the output from the linear layer $f(\boldsymbol{x})$ by multiplying $T$ to $f(\boldsymbol{x})$ as $Tf(\boldsymbol{x})$. $T$ plays the role of re-weighting $f(\boldsymbol{x})$, making the model give different confidentiality to the clean label and noisy label and also adapt to the noise. The optimization goal is to optimize classifier $f$ that minimizes Eq. 4.

## 5 Experiment

## 5.1 Experimental Setting

We implement our approach with PyTorch [24] 1.8.0 and Torchvision 0.9.0 with 2 NVIDIA 2080Ti GPUs. The models are trained using Adam optimizer [14] by setting coefficients of Adam $\beta_1 = 0.9, \beta_2 = 0.999$, and $\epsilon = 10^{-8}$. The learning rate is 0.0001 and remains constant during the whole training process. The batch size and total epochs are set to 64 and 50, respectively. In addition, we set the warmup step $\tau = 10$ as the default value.

## 5.2 Dataset

*5.2.1 ISIC Skin Dataset.* The ISIC skin dataset [2] comes from the Melanoma Project initiated by the International Skin Imaging Collaboration. There are 2600 melanoma dermoscopy images in this dataset, classifying malignant and benign classes. The dataset consists of 2000 melanoma dermoscopy images for training and 600 for testing. In addition to these data, we also utilize an extra 1582 images from the ISIC training archives for training. All images are resized to $224 \times 224$, and are augmented by randomly adopting horizontal flipping, vertical flipping, or image rotation.

**Table 1: Comparison of classification results with state-of-the-art methods on the ISIC and the HCDD dataset (Accuracy, %). *Noise* here refers to the noisy sample ratio in the whole dataset. The results are presented in `Mean(std)` format. The bold numbers are the best result among all the methods.**

| Dataset | Noise | Cross entropy | Mentornet | Co-teaching | ELR | JoCoR | JoCoR+NIB | RMIC | Ours |
|---------|-------|---------------|-----------|-------------|-----|-------|-----------|------|------|
| ISIC | 0.05 | 84.25(0.66) | 83.47(0.27) | 83.98(0.45) | 84.28(0.23) | 68.78(1.41) | 69.92(1.91) | 85.40(0.22) | *87.79(0.55)* |
| | 0.1 | 83.01(0.37) | 83.83(0.37) | 83.39(0.50) | 83.18(0.35) | 64.41(2.31) | 66.74(0.81) | 84.33(0.31) | *86.39(0.38)* |
| | 0.2 | 81.36(0.61) | 82.15(0.49) | 83.22(0.21) | 83.37(0.48) | 60.78(3.80) | 62.05(1.13) | 84.17(0.32) | *85.39(0.26)* |
| | 0.4 | 69.65(0.65) | 70.40(0.33) | 74.78(0.44) | 72.98(0.26) | 52.02(3.25) | 53.07(0.16) | 76.67(0.14) | *79.17(0.58)* |
| HCDD | 0.05 | 91.28(0.57) | 92.65(0.32) | 92.88(0.36) | 92.69(0.33) | 88.85(1.89) | 90.46(0.33) | 93.88(0.35) | *94.11(0.27)* |
| | 0.1 | 88.95(0.31) | 91.05(0.27) | 91.39(0.48) | 90.84(0.24) | 86.58(1.92) | 87.34(1.09) | **92.75(0.30)** | *92.63(0.36)* |
| | 0.2 | 82.67(0.49) | 85.66(0.27) | 86.05(0.17) | 87.12(0.28) | 86.41(1.61) | 84.95(1.60) | 90.88(0.20) | *91.83(0.38)* |
| | 0.4 | 63.41(0.63) | 69.43(0.47) | 76.03(0.18) | 73.86(0.17) | 75.81(2.29) | 74.98(3.12) | 79.00(0.34) | *80.48(0.26)* |

**Table 2: The accuracy of different methods on NIH chest X-ray datasets. The average accuracy and standard deviation of 5 random runs are reported and the best results are in bold. The results are presented in Mean(std) format.**

| Method | Pneumothorax | Nodule or Mass |
|--------|--------------|----------------|
| Cross entropy | 0.870(0.33) | 0.843(0.39) |
| Mentornet | 0.866(0.27) | 0.837(0.31) |
| Co-teaching | 0.873(0.21) | 0.820(0.15) |
| ELR | 0.871(0.23) | 0.832(0.21) |
| RMIC | 0.891(0.14) | 0.846(0.22) |
| Ours | **0.895(0.21)** | **0.851(0.14)** |

*5.2.2 Kaggle Histopathologic Cancer Detection Dataset.* The Kaggle histopathologic cancer detection dataset (HCDD) is a marginally modified version of the PatchCamelyon (PCam) [1, 32]. The dataset is a binary classification for malignant and benign, containing low-resolution images of lymph node sections with about 30, 0000 markers extracted from digital histopathological scans. We randomly select 6200 and 800 images from the training images as our training data and test data, respectively. All the images are resized to the size of $224 \times 224$.

*5.2.3 NIH Chest X-Ray Dataset.* This dataset [33] contains 112,120 frontal-view CXR images from 32,717 patients. Each image is labeled with 14 possible pathological findings that are automatically mined from the text reports. As clean testing data, we employed 1,962 manually labeled images [20]. We resized all the skin images to the size of $224 \times 224$ and normalized each image by subtracting the ImageNet mean and std.

## 5.3 Comparison of Classification Results With State-of-the-Art Methods

The performance of our method under different unsupervised loss weights on the histopathologic datasets. To demonstrate the effectiveness of our algorithm, we compare `Pseudo-T correction` with five state-of-the-art noisy label learning methods, including:

*(1) MentorNet (2018) [11].* MentorNet proposes a neural network to supervise the training of the base deep networks. They used a group of clean data to weigh each training data. This method learns a data-driven curriculum dynamically with the network.

*(2) Co-teaching (2018) [8].* This method trains two deep neural networks to teach each other given every mini-batch by selecting some data of possibly clean labels. Secondly, the two networks communicate with each other about what data should be used for training. Then, each network updates itself and back-propagates the data chosen by its partner networks.

*(3) ELR (2020) [19].* A semi-supervised learning technique to generate targe probabilities based on the model outputs. Then, ELR develops a regularization term that indirectly prevents memorizing the erroneous labels by guiding the model towards these targets.

*(4)JoCoR (2020) [35].* A learning paradigm used in a co-training network structure. It considers and trains the two networks as a whole, and aims to reduce their diversity. The two networks first make predictions on the same mini-batch and calculate the joint loss to maximize the agreement between the two classifiers. The small-loss examples are then selected to update the two networks simultaneously; consequently, the two networks become similar.

*(5) NIB (2021) [40].* A plugin module that separates hard samples from mislabelled samples to further increase the network's performance under a noisy environment. The module consists of a probability transition matrix to generate accumulative soft labels. The divergence between the soft labels and the network's predictions is calculated. With less divergence, the hard samples are identified. Zhang *et al.* prove that NIB brings improvement to the JoCoR framework with experiments.

*(6) RMIC (2022) [38].* A collaborative training paradigm with global and local representation learning. They use a self-ensemble model with a noisy label filter to select the clean and noisy samples. Then, the clean samples are trained and applied to eliminate the disturbance from imperfect labeled samples. This method uses a teacher-student co-training strategy for local and global knowledge distillation.

*5.3.1 Results on ISIC Dataset.* Since the data is more imbalanced and higher proportion of hard samples than the lymph node classification task, it should be noted that the skin lesion classification data is more challenging to use. In the malignant vs. benign classification task on the ISIC dataset, `Pseudo-T correction` significantly outperformed the alternative methods. From Table 1, it is observed that when there is a 0.05 noise ratio setting, the poor sample utilization

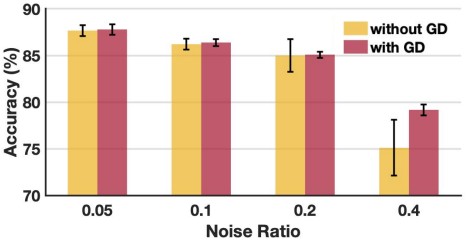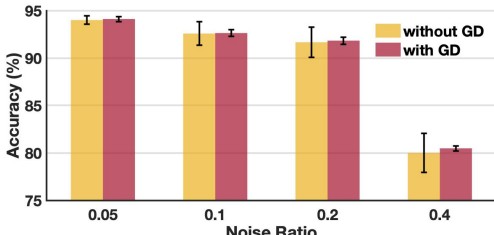

**Figure 3: Ablation study on stopping criteria (GD). Left: ISIC, Right: HCDD.**

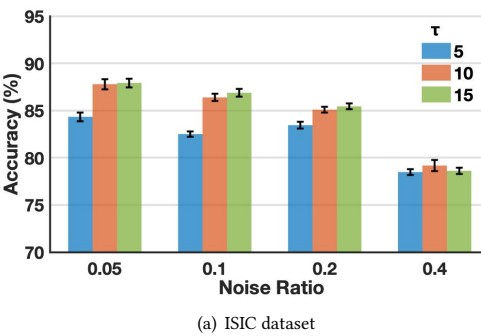

(a) ISIC dataset

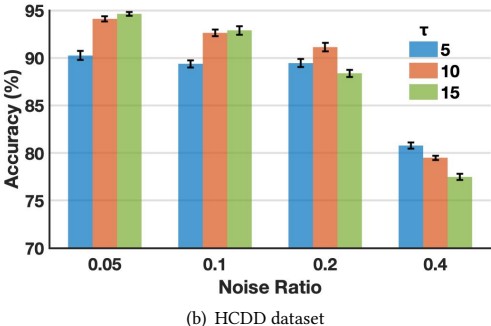

(b) HCDD dataset

**Figure 4: The performance of ablation study under different warmup epochs ($\tau$) and noise ratio on ISIC dataset and HCDD dataset. We report the testing accuracy under distinct settings with the format of mean and std.**

**Table 3: Classification results of with vs. without `Pseudo-T correction` on ISIC dataset (Accuracy, %). *Noise* here refers to the noisy sample ratio in the whole dataset. The results are presented in `Mean(std)` format.**

| `Pseudo-T correction` | Noise | ACC (std) | `Pseudo-T correction` | Noise | ACC (std) |
|---|---|---|---|---|---|
| | 0 | 88.74 (1.23) | | 0 | 87.73 (1.10) |
| | 0.05 | 87.64 (0.12) | | 0.05 | 87.53 (1.03) |
| w | 0.1 | 86.88 (1.12) | w/o | 0.1 | 86.53 (1.24) |
| | 0.2 | 84.66 (0.96) | | 0.2 | 84.53 (1.61) |
| | 0.4 | 81.11 (1.67) | | 0.4 | 76.41 (2.62) |

rate and incorrect hard sample identification in the mild scenario make co-teaching ineffective, while the biased sample selection and down-weighted hard samples in the self-paced Mentornet model hindered its effectiveness. The RMIC discovered a slight increase in accuracy when the noise ratio was low, but when the noise ratio approached 0.4, the performance drastically declined because their cross-entropy loss on small loss data is not reliable in this situation. Compared with our methods, the RMIC led to a minimal difference in accuracy between decentralized training and the baseline, using a noisy dataset led to a much larger difference between the two. The results show that our method can successfully handle noisy labels since the transition matrix can effectively utilize the hard samples and the noise label filter can filter out the noise labels.

*5.3.2 Results on HCDD Dataset.* To evaluate the generalization ability, Table 1 also presents the testing accuracy (summarized from three independent runs) of `Pseudo-T correction` and comparison methods on different noise levels. Although the RMIC method achieves the best, 92.75% value at noise equals 0.1 on the HCDD

dataset, this method is based on a co-training strategy, so it may be challenging to employ their cross-entropy loss on small loss data. This plausibly implies that `Pseudo-T correction` would be very helpful and practical in clinical applications by fully using the information of noise data.

In addition to successfully handling label noise, our method is lightweight and easy to implement on resource-limited devices. While using ResNet18 as our base model, `Pseudo-T correction` contains 11.2 million trainable parameters, which is only 1/4 that of RMIC and Co-teaching and around half of the other methods. Compared to RMIC [38], our method gains comparable results to RMIC while being simple and provable.

*5.3.3 Results on NIH Chest X-Ray Dataset.* The NIH Chest X-Ray dataset is a real-world dataset with real noise. We present the comparison in test accuracy with state-of-the-art methods on the NIH chest X-ray dataset in Table 2. From this Table 2, we can observe that `Pseudo-T correction` still outperformed the baseline (*e.g.*, Cross entropy and Co-teaching) and other methods. For instance, our

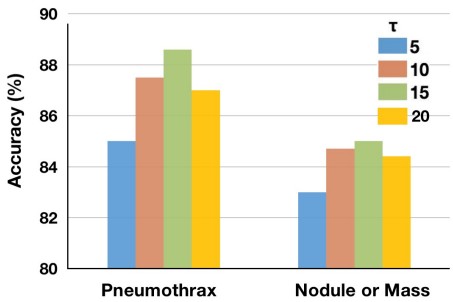

**Figure 5: The performance of ablation study under different warmup epochs ($\tau$) and noise ratio on NIH chest X-ray dataset. We report the testing accuracy under distinct settings.**

method achieves 2.4% accuracy gains over ELR. Compared to RMIC, our method achieves better performance with 0.5% accuracy-score gains. We explain this to be that the sample selection is difficult because of the tiny noise proportion and unbalanced sample distribution of the 14 diseases. Overall, this experiment shows the robustness of our method in a real clinical context. Thus, our method is practical with considerably better adaptability.

## 5.4 Ablation Study

*5.4.1 Sanity Check on the Estimated Transition Matrix $T$.* The transition matrix $T$ is an essential element in improving the classification performance. We conducted multiple experiments to prove the correctness of the transition matrix. For sanity check, we visualize the convergence of transition matrix $T$ on the ISIC dataset when the noisy ratio is 0.05 and 0.1. The investigations of $T$ are presented in Table 3. As shown in Table 3, our method achieves ACC of 0.1 noisy ratio gains over 0.35% compared to the without Pseudo-T correction. In addition, compared to the without transition matrix correction under a 0.4 noisy ratio, **Pseudo-T correction** added after it achieves significant performance improvement, especially with the increase of 4.7% ACC. Experimental results indicate that the minimization of the corrected loss under the noise distribution corrected by the correction matrix is the same as the minimization of the original loss under the true label distribution. It demonstrates the advantages of using **Pseudo-T correction** to reduce the risk of training a classification model with incorrect information because the chances of randomly selecting a complementing label that is not a real label are high.

*5.4.2 Influence of Stopping Criteria (GD).* To evaluate the stopping criteria (GD) on the ISIC dataset and the HCDD dataset, we report the results in Fig. 3. It can be seen from Fig. 3 that the proposed **Pseudo-T correction** produces overall good results under a distinct noise ratio. As shown in Fig. 3, we report the comparison results with vs. without stopping criteria (GD). From Fig. 3, the "with GD" achieves the improve- ment of 4.07% compared with "without GD" under 0.4 noise rate on the ISIC dataset. Equally, under the noise rate is 0.05, our method achieves a good result with GD. At a noise rate of 0.4, "with GD" increases the accuracy over "without GD" by 1.3% on the HCDD dataset. Fig. 3 suggests that our

method using stopping criteria produces the overall better performance, compared with that without stopping criteria on ISIC and HCDD datasets. This means that our method would be very helpful in noisy data by using gradient disparity as an effective criterion to avoid overfitting noisy data with iterative methods.

*5.4.3 Influence of Warmup Epoch.* We evaluate the influence of warmup epoch, by comparing the results of **Pseudo-T correction** using different pre-training epochs under the settings of different noise ratios. Specifically, we vary the noise ratio from the range of [0.05, 0.1, 0.2, 0.4] and record the results in Fig. 4. It can be seen from Fig. 4(a) that, with a very low warmup epoch (e.g., $\tau = 5$), our method cannot yield good results. With $\tau = 10$, we can obtain relatively better results. The results are reported in Fig. 4(a), **Pseudo-T correction** using 0.05 noisy ratio and 15 warmup epochs to produce the best performance on the ISIC dataset. The main reason could be that the warmup epoch helps extract robust features of noisy samples.

Similarly, it can be seen from Fig. 4(b) that our method yields consistently better results within the range of [0.05, 0.1], when $\tau$ is equal to 15. This Fig. 4(b) also shows that, with $\tau = 10$, **Pseudo-T correction** achieves the overall best performances under a noise ratio of 0.2. Also, these results imply that the warmup epoch is an important parameter for maintaining model stability.

We further analyze how the warmup epoch $\tau$ on NIH chest X-ray dataset. We ranged $\tau \in [5, 10, 15, 20]$. As shown in Fig. 5, we can observe that the best selection of $\tau$ is set to 15. From 5 to 15, increasing $\tau$ can improve the classification performance. The performance slightly drops when $\tau$ is set to 20. Thus, we report the performance of the proposed method by setting $\tau$ to 20 on NIH chest X-ray dataset, unless otherwise specified. We find that increasing the $\tau$ may not always improve the performance yet brings extra computational cost. It also demonstrates that increasing $\tau$ is not always the best choice for architectural design.

## 6 Conclusion

In this work, we present a simple and provable method by utilizing noisy labels for medical image analysis. The proposed method does not rely on relabeling the noisy labeled data and directly uses noisy labeled data and a lightweight network to promote the learning of robust representation features. Notably, to tackle the realistic but ignored issue of incomplete information in noisy labels, we designed a transition matrix, that corrects model predictions. Furthermore, to avoid underfitting or overfitting deep neural networks trained with iterative methods, resulting in inaccurate transition matrix estimation. Therefore, we employ gradient disparity as a criterion to decide if fixing the transition matrix in the further model training steps. Extensive experiments on three challenging medical datasets, including dermoscopic images, histopathology slide images, X-Ray, about medical image classification tasks with random noise and inter-observer variability, demonstrate that our method obtains state-of-the-art performance. In the future, we will fuse multi-class classification tasks to address the data heterogeneity issue and apply our methods to further improve performance on more datasets.

# Acknowledgments

The authors gratefully acknowledge the support from the following foundations: National Natural Science Foundation of China under Grant 62322309, Shanghai Science and Technology Innovation Action Plan under Grant 23S41900500, Natural Sciences and Engineering Research Council of Canada (NSERC), Compute Canada, and Vector Institute.

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
