# OpenReview forum: "A Simple and Provable Approach for Learning on Noisy Labeled Multi-modal Medical Images"
_acmmm.org/ACMMM/2024/Conference — MM2024 Poster_

### Official Review · Reviewer_aj4m · 2024-05-22

**Rating:** 2
**Confidence:** 4

**Summary:**

This paper focuses on noisy label learning for medical image classification. The main components are label transition matrix estimation and label correction. The framework is easy to follow. Extensive experiments and ablation studies on three public datasets have been conducted for evaluation.

**Strengths:**

Noisy label learning is of great value in clinical applications. The paper is easy to follow.

**Limitations:**

1. Nothing related to multi-modal medical images. It is inconsistent with the title.
2. Limited innovation. Both noisy label identification and label correction follow standard operations. The main content namely estimating the transition matrix largely depends on noisy label identification.
3. Simple noise settings. Random flipping is not realistic.
4. Following comment 3, only binary classification is too simple. Noisy multi-label learning is more interesting and more realistic noisy settings are expected. For instance, instance-dependent noise.
5. Insufficient comparison. More advanced noisy label identification methods should be included for comparison.
6. Experiments on multi-label classification are needed.

**Suitability:**

2

---

### Official Review · Reviewer_T8N7 · 2024-05-24

**Rating:** 3
**Confidence:** 3

**Summary:**

This paper proposes a novel method for medical image analysis, i.e., Pseudo-T correction, to leverage the labels of clean and noisy data. Pseudo-T correction first splits clean and noisy samples via a noisy label filter, and enhances forwarding corrction by taking advantage of transition matrix estimation to fit the observed labels. In the given experimental settings and with the three medical image datasets with noisy labels, the proposed model shows improvements over SOTA methods.

**Strengths:**

1）The proposed Pseudo-T correction is technically sound by estimating the transition matrix and updating label probabilistic modeling to learn discriminative representation features.

2）Experiments on three real-world datasets verify the effectiveness and practicability of the proposed Pseudo-T correction method.

3）This paper may inspire significant practical applications to medical image analysis tasks, in terms of efficiency and universality of noisy label training.

**Limitations:**

1) The paper is an incremental method so that the novelty of this proposed method is limited, which estimates a transition matrix to correct model outputs and then employs gradient disparity as a criterion to assign different confidences to the clean and noisy labels in the further model training steps.
2) It is a bit unclear of the motivation behind the technical design of the proposed model. It is claimed that this paper finds that most existing heuristics only detect and utilize single-modal clean label (Line 134). However, I am confused with how this paper solve the “single-modal” dilemma, and provide accurate predictions on multi-modal medical images.
3) It could be more convincing to vary the noise rate from 0 to 0.9 to demonstrate the robustness to noise
4) Figures can be further improved for better readability. For example, in Figure 2, for noisy data (green) and clean data (yellow), the explanation of the circles framed by red lines is missing.
5) The presentation of this paper needs improvement:

        Line 134: Nevertheless,... -> Moreover,...
        Line 154: ...three medical image noisy labels with public datasets -> ...three public medical image datasets with noisy labels
        Line 267: K-Mean Clustering -> K-Means Clustering
        Line 384-389: The label of the loss minimizer fomulation is missing
        Line 390: The numbering and notation in the formulas are confusing and some marks are missing. Clear and consistent notation is essential for readability and understanding.
        Line 410: The indentation in the flowchart of Algorithm 1 is difficult to follow. What is the output of the training process?

**Suitability:**

2

---

### Official Review · Reviewer_d2Lk · 2024-05-29

**Rating:** 5
**Confidence:** 4

**Summary:**

The Pseudo-T correction method provides a simple, yet effective, approach to handle noisy labels in medical image classification. It leverages a transition matrix to correct predictions and employs gradient disparity to prevent overfitting. The method demonstrates superior performance on multiple datasets and is computationally efficient, making it a valuable tool for clinical applications.

**Strengths:**

1. The method presents a novel and theoretically sound approach to a significant problem in medical image classification.
2. Experimental results show strong performance improvements and robustness across multiple datasets.
3. The paper is clear and well-structured, making the methodology accessible.

**Limitations:**

1. Limited novelty in broader context, as the use of transition matrices for label correction is not entirely new.
2. More comprehensive comparisons with additional recent approaches[1][2] would strengthen the validation.
[1] Hongxin Wei, Lei Feng, Xiangyu Chen, and Bo An. Combating noisy labels by agreement: A joint training method with co-regularization.
[2] Zhang, Z., Li, Y., Wei, H., Ma, K., Xu, T., & Zheng, Y. Alleviating noisy-label effects in image classification via probability transition matrix.

3. Including real-world implementation examples would enhance the practical relevance.

**Suitability:**

3

---

### Meta-Review · Area_Chair_jbz8 · 2024-06-30

**Recommendation:** Accept (Poster)
**Confidence:** 4

**Metareview:**

This paper works on noisy label learning for medical image classification. The rebuttal partially addressed the reviewers’ concerns, and this paper was rated as 1 weak accept, 1 borderline accept and 1 borderline reject. Although some concerns on the technical novelty remains, I tend to agree more on the merits of this paper on simple and provable approach for noisy label learning. Thus, I tend to recommend accept after reconciling the reviewers’ comments. I do suggest the authors revise the manuscript significantly by taking all the reviewers’ comments into consideration.